# Hypertension Induces Pro-arrhythmic Cardiac Connexome Disorders: Protective Effects of Treatment

**DOI:** 10.3390/biom13020330

**Published:** 2023-02-09

**Authors:** Matus Sykora, Katarina Andelova, Barbara Szeiffova Bacova, Tamara Egan Benova, Adriana Martiskova, Vladimir Knezl, Narcis Tribulova

**Affiliations:** Centre of Experimental Medicine, Slovak Academy of Sciences, Institute for Heart Research, 84104 Bratislava, Slovakia

**Keywords:** primary and pulmonary hypertension, connexome, cardiac arrhythmias and treatment

## Abstract

Prolonged population aging and unhealthy lifestyles contribute to the progressive prevalence of arterial hypertension. This is accompanied by low-grade inflammation and over time results in heart dysfunction and failure. Hypertension-induced myocardial structural and ion channel remodeling facilitates the development of both atrial and ventricular fibrillation, and these increase the risk of stroke and sudden death. Herein, we elucidate hypertension-induced impairment of “connexome” cardiomyocyte junctions. This complex ensures cell-to-cell adhesion and coupling for electrical and molecular signal propagation. Connexome dysfunction can be a key factor in promoting the occurrence of both cardiac arrhythmias and heart failure. However, the available literature indicates that arterial hypertension treatment can hamper myocardial structural remodeling, hypertrophy and/or fibrosis, and preserve connexome function. This suggests the pleiotropic effects of antihypertensive agents, including anti-inflammatory. Therefore, further research is required to identify specific molecular targets and pathways that will protect connexomes, and it is also necessary to develop new approaches to maintain heart function in patients suffering from primary or pulmonary arterial hypertension.

## 1. Introduction

Herein, we focus on pro-arrhythmic disorders elicited by primary hypertension (HTN). This is a prevalent risk factor for cardiovascular disease in the general population that includes younger people. HTN is defined as systolic blood pressure above 140 mmHg or diastolic blood pressure above 80 mmHg. These can result in hypertensive heart disease from pressure overload. HTN-induced compensated myocardial hypertrophy can aggravate left heart dysfunction over an extended period. This is mainly due to fibrosis, and it results in heart failure (HF). HTN also promotes the occurrence of cardiac arrhythmias, including ventricular fibrillation (VF) and atrial fibrillation (AF). In addition to HTN, there is also an increased incidence of pulmonary arterial hypertension (PAH) which affects right heart function. PAH is caused by pulmonary vasculopathy, and it results in elevated pulmonary arterial pressure above 25 mmHg at rest. This is a most serious clinical problem and has adverse prognosis due to heart disease progression and the accompanying myocardial fibrosis contributing to HF. There is also a further risk of VF and AF. Further research is also essential to differentiate gender differences in both primary HTN and PAH for efficient treatment.

It has been established that myocardial connexin-43 (Cx43) channels ensure electrical coupling between cardiomyocytes and are crucial in the development of malignant cardiac arrhythmias [1]. The available literature indicates that HTN and PAH deteriorate Cx43 channels and mediate communication at the heart’s gap junctions (GJs), as well as the function of adhesive junctions [2,3,4,5,6]. Cx43 GJs are essential for electrical coupling and the propagation of action potentials between cardiomyocytes [7]. Desmosomes (D) and adherens junctions (AJs) are junctions responsible for cardiac myocyte adhesion and mechanical force transduction via actin filaments [8] located in the intercalated discs (ID), and their impairment or dysfunction in heart disease can facilitate VF or AF and contribute to HF [9,10,11].

Research suggests that it is important to consider connexomes and “area composite”, complex proteins which underly the interaction of Cx43-formed GJs, D, and AJs at the intercalated discs. This is highlighted in Figure 1, and Figure 2 shows their localization on the cardiomyocyte lateral aspect. Connexome defects are essential in arrhythmogenic cardiomyopathy [12,13,14,15,16,17], and are most likely implicated in both AF and VF promoted by HTN or PAH [11,18,19].

It is most important that the connexomes affect both cardiomyocyte electrical coupling through Cx43 channels and mechanical force transduction by adhesive junctions. It is *conditio sine qua non* for synchronized myocardial contraction, and this is impaired in heart disease [8,9,20]. In addition, ion channels and Cx43 hemichannels form part of the connexome, and these affect its function [7]. Therefore, connexome’s structural and functional preservation presents a challenging target for pharmacological and non-pharmacological approaches [10]. These will enhance the fight against life-threatening cardiac arrhythmias in general, and in HTN and PAH in particular.

**Figure 2 biomolecules-13-00330-f002:**
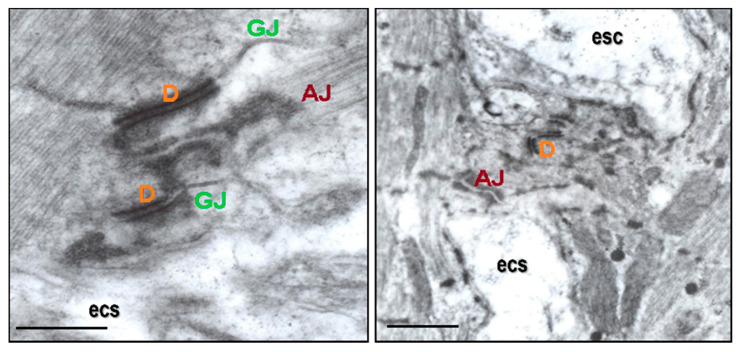
Connexomes are identified on the lateral sides of the hypertrophied guinea pig cardiomyocytes. Their AJ, D, and GJ components are destroyed by progressive extracellular collagen deposition [21]. Scale bar represents 1 micrometer.

## 2. Factors Involved in the Development of Re-Entrant Cardiac Arrhythmias, VF, and AF

The heart can “die” due to the following three major events: electromechanical dissociation, asystole, and VF, which is the most frequent case. In addition, AF is the most frequent arrhythmia in the population. AF deteriorates heart function and can cause stroke. The basic electrophysiological mechanisms of cardiac arrhythmias include the following: (1) abnormal electrical impulse generation through ectopic pacemaker-like activity or triggered activity and (2) abnormal electrical impulse propagation, due to blocks of conduction and re-entrant excitation. Simultaneous operation of abnormalities 1 and 2 may occur [7,22,23,24,25].

Figure 3 highlights that arrhythmogenic substrates, electrical triggers, and modulations are the three major factors in AF and VF development. Heart disease-related myocardial structural and ion channel remodeling are the established arrhythmogenic substrates. This includes hypertrophy, fibrosis, D and AJ impairment, and altered Cx43 topology and its downregulation. These changes can influence anisotropic conduction and promote discontinuous and re-entrant action potential propagation [26,27]. Fibrosis causes extreme disturbance of Cx43 GJ distribution at the myocardial interface, and defines the location of re-entry circuits that cause ventricular arrhythmia [2]. In addition, heterogeneous Cx43 GJ expression adversely affects the normal pattern of coordinated myocardial excitation, and this can directly depress cardiac performance [28].

Research indicates that abnormal Ca^2+^ handling, Ca^2+^-overload, and ischemia-related acidosis induce Cx43-GJ uncoupling. When this is combined with ion current abnormalities, it can trigger the electrical disorders that initiate AF or VF occurrence [4,6,19,24,29,30].

Modulating factors, such as humoral and autonomic tone misbalance [31,32], inflammation, and redox dysregulation [22,33], as well as various stressors, such as stretch [34,35], impact the susceptibility of the heart to re-entrant arrhythmias. Finally, Cx43 hemichannel activation is fundamental to this process, because it can promote pro-arrhythmic signaling [7,24] and Cx43 GJ redistribution or inhibition.

Although the basic mechanisms that can cause cardiac arrhythmias are known, there is little understanding of the changes in cardiac electrical properties in heart disease, and this includes those in HTN and PAH. These properties appear to be the immediate cause of the operation of the arrhythmogenic mechanisms which occur in cardiac conditions and disease. Both AF and VF are assumed to occur due to ectopic impulse initiation, blocking conduction, and circuit re-entry. Although, AF can self-terminate and VF can be transient in the heart, avoiding arrhythmogenic substrates under the control of modulating factors. However, sustained AF or VF can aggravate the myocardial injury and connexome function (Figure 4). Therefore, protection of the connexome, GJ, and adhesion of the perinexus to preserve sodium channels and cardiac conduction present promising anti-arrhythmic research targets [36].

## 3. Connexome Impairment Promoting AF or VF Development in Primary HTN

Primary or essential HTN is a prevalent risk factor jeopardizing cardiovascular health, and its increase in younger people is very alarming [37,38]. HTN is a multifactorial disease, and its etiopathogenesis includes the interplay between the genetic, epigenetic, environmental, and lifestyle factors which lead to meta-inflammation, oxidative stress, and auto-antibody production [23,39,40,41,42,43,44,45,46,47,48,49,50,51,52]. In addition, insufficient myocardial perfusion to match overall metabolic demand has been identified as an elevated risk of heart failure (HF) in symptomatic patients with HTN [53]. Adverse HTN consequences promote stroke, as well as heart electro-mechanical dysfunction, and this can lead to the development of AF or VF and subsequent HF [54,55,56,57]. Therefore, primary or essential HTN has become a target of considerable global public health concern [38,58].

Numerous animal and human heart studies indicate that HTN-induced structural remodeling involves hypertrophy, a shift in myosin heavy chains, cytoskeletal proteins, fibrosis, channelopathy, altered Ca^2+^ handling, and Cx43 disorders of the left ventricle. These are crucial in impaired conduction, electrical instability, heart mechanical dysfunction, and cardiac arrhythmias [47,54,55,57,59,60,61,62,63,64,65,66,67,68,69,70].

In addition, the progression of left ventricular alterations to HF is associated with right ventricular dysfunction and electrical instability [71]. This contributes to an adverse prognosis. Moreover, changes in membrane lipid composition have been reported in cardiac hypertrophy, and these can also affect GJ coupling and conduction [72].

Available literature indicates that HTN results in altered connexomes on the cardiomyocyte lateral sides, in addition to its multiple locations in ID. The subcellular alterations indicate connexome abnormalities. These include the AJ dehiscence, internalization, and GJ degradation, as demonstrated in Figure 5 and Figure 6.

HTN-induced structural remodeling causes Cx43 redistribution from GJs to the cardiomyocyte lateral sides [61,62,67,74], Cx43 downregulation [73,75,76,77], and D and AJs dehiscence [4,19]. Delocalization of the Cx43 from GJs is a major pathologic-mechanism in hypertensive electrochemical remodeling [74,78]. The N-cadherin/catenin complex of AJs is a master regulator of ID function [18], and N-cadherin loss leads to altered Cx43 with reduced conduction velocity (CV) and arrhythmogenesis [13,39].

It is important that Cx43 cardiomyocyte lateralization is accompanied by the remodeling of D and microtubule-associated proteins [79]. This remodeling can affect electrical synchrony under conditions of disrupted ID integrity. Cadherin dysregulation has been demonstrated in IDs of spontaneously hypertensive stroke-prone rats [80]. This can contribute to altered heart function. In addition, reduced Cx43 expression triggers increased fibrosis due to enhanced fibroblast activity [81]. Therefore, the implications of arrhythmogenic fibroblast–cardiomyocyte interactions should be considered [82].

LVH is associated with increased intracellular resistivity which can be solely attributed to increased junctional resistance between adjacent cells [83]. Cardiac hypertrophy is known to be regulated by micro RNAs [84], and the upregulation of muscle-specific miR-1 noted in hypertrophy can affect cardiac arrhythmogenicity by targeting the GJA1 gene which encodes Cx43 [85]. Gender differences in miR-1 [86] can partly explain higher Cx43 protein levels in females [87] and their lower cardiac arrhythmia susceptibility [55]. It is also important that chronic distress promotes miR-1 expression [35], and this increases Cx43 displacement and induces ventricular tachyarrhythmias in hypertrophic rat hearts [88]. In addition, there is dysregulated miR-1 processing in the SHR heart associated with aging [89]. The implication of miRs in the development of cardiac arrhythmias has also been comprehensively reviewed [90].

GJ remodeling in human decompensated cardiac hypertrophy is associated with increased interaction of Cx43 with zonula occludens-1 (ZO-1) [91,92]. This could be implicated in the downregulation and decreased size of Cx43 GJs contributing to the arrhythmic substrate. In addition, ZO-1 as a connexin-interacting protein, determines AJ and GJ localization at the intercalated disc [93]. Moreover, high mechanical load induces rapid Cx43 phosphorylation loss, followed by decreased Cx43 protein levels [94]. It has been revealed that phospho-Ser-368 Cx43 channels were segregated into the GJ center following PKC activation, and these were subsequently internalized and degraded [95]. Of note, ubiquitination is critical for GJ internalization [96], desmin mediates TNF-α-induced aggregate formation, and ID reorganization in the failing heart [97].

Data in the literature indicate proposed factors and mechanisms that may be involved in pro-arrhythmic “connexome” dysfunction in chronic arterial hypertension, as outlined in Figure 7.

### Interventions Associated with Protecting Connexome in Primary HTN

The spontaneously hypertensive rat (SHR) is the most frequent experimental model used to imitate human primary HTN. The SHR treatment with the phenylbutyrate short-chain fatty acid derivative and captopril both improved myocardial function, regressed cardiac hypertrophy, and enhanced recovery from HF [98]. It is important here that the gene set combination related to oxidative stress, growth, inflammation, protein degradation, and pro-fibrotic TGF-β signaling were downregulated; and these effects were most likely associated with improved connexome function.

The literature cites the following treatment influences on cellular communication, impulse conduction, and adhesive junctions:

(1) The distribution of Cx43 GJs became more regular and confined to the ID and attenuation of LVH in SHR after treatment with the atorvastatin 3-hydroxy-3-methylglutaryl coenzyme A inhibitor [99]. It is also noted [22] that statins have pleiotropic anti-inflammatory and anti-oxidative effects, and that they alter membrane lipid composition. This could protect connexome function.

(2) The activation of Ang II and AT1 receptors decreases GJ conductance in the failing heart, with consequent impairment of impulse propagation [100]. In addition, both Ang II and aldosterone promote inflammation and enhance collagen deposition and interstitial fibrosis with serious consequences for the spread of electrical activity through the myocardium. Ang II also induces sudden arrhythmic death and electrical remodeling in rats which harbor the human renin and angiotensin genes [68].

(3) The arrhythmias was attributed to inflammation, interstitial fibrosis, reduced transcripts of potassium channel subunit Kv4.3, and gap-junction Cx43 that was partly abolished by losartan. However, a chronic losartan blockade of the Ang II AT1 receptors increased intercellular communication, reduced fibrosis, and improved impulse propagation in the failing heart [100]. Ventricular conductance velocity was also enhanced to some extent by increased GJ conductance, decreased interstitial fibrosis, and structural remodeling.

(4) Candesartan is a further receptor blocker, but its action did not cause fibrosis regression in the SHR’s left atrium at a dose sufficient to reduce blood pressure and left ventricular hypertrophy [101]. In contrast to Ang II, angiotensin (1–7) has an opposite effect on impulse propagation, excitability, and cardiac arrhythmia [102].

(5) Gap junction A1-20k is required for Cx43 passage to the intercalated disc. This attenuates LVH by regulating GJ formation and mitochondrial function [103].

(6) Resolvin D2 protected cardiovascular function and structure when administered before and after the development of Ang II-induced HTN by attenuating inflammation and fibrosis [104]. It is also important that hydrogen sulfide attenuated inflammation by regulating lymphocyte-confined Cx43 expression in the SHR [105]. This indicates that the resolution of inflammation could be an effective therapy against the target organ damage associated with HTN.

(7) This concept is supported by recent reviews which stressed the pleiotropic antiarrhythmic properties of cardioprotective agents affecting inflammation and oxidative stress [22,106]. These included statin sodium glucose co-transporter-2 inhibitors (SGLT2i) and omega-3 fatty acids. Consequently, these compounds attenuated the downregulation of myocardial Cx43 expression and its abnormal topology, and reduced fibrotic areas in heart ventricles in conditions, such as primary HTN.

(8) However, defective fatty acid uptake is involved in myocardial remodeling in the SHR [107]. In contrast, omega-3 fatty acids intake attenuated Cx43 GJ lateralization and ameliorated the integrity of GJs, AJs, and the sarcolemma in SHR hearts. This rendered them less susceptible to inducible VF [40,55,73,76].

(9) Omega-3 fatty acids decreased the protein kinase PKCδ involved in SHR extracellular matrix remodeling [108]. These acids also increased the PKCε expression associated with Cx43 preservation at the GJs [109].

(10) Dietary omega-3 fatty acids and renin inhibition with aliskiren improved both electrical remodeling and the antiarrhythmic effects attributed to improved Cx43 expression. This also prevented Cx43 redistribution in the model of high human renin primary hypertension [69]. In addition, aliskiren ameliorates maladaptive Cx43 remodeling in the SHR [110].

(11) Finally, early primary HTN therapy can attenuate myocardial structural remodeling and suppress myocardial LVH and fibrosis in the SHR [111]. For example, relaxin can suppress AF by reversing fibrosis and LVH, and it can then increase CV and Na+ current in this rat strain [112].

## 4. Connexome Impairment Promoting VF or AF in PAH

There is an increasing incidence of pulmonary arterial hypertension (PAH) in the general population. This is in addition to problems caused by primary HTN. The elevated pressure in PAH affects the pulmonary circulation and right heart function, and the attendant proliferative vasculopathy results in ongoing increased right ventricular afterload, structural remodeling, and mechanical HF. PAH is multi-factorial, and its etiopathogenesis is not completely understood because of its relationship with underlying somatic disease [49].

PAH is similar to primary HTN, as it is influenced by genetic, epigenetic, and environmental factors. However, hypertension is the most common causative factor of cardiac remodeling [113,114]. PAH also has similar involvement in the renin-angiotensin-aldosterone signaling system, and noncoding RNAs have a prevalent effect as biomarkers and therapeutic targets in preventing heart dysfunction and malignant cardiac arrhythmias [49,115].

Various authors consider that Cx43 downregulation, its dephosphorylation and internalization, and the dysregulated Cx43-mediated signaling in the right ventricle offer crucial intervention targets. [116,117,118,119]. In addition, Cx43 heterogeneous expression in the right ventricular outflow tract presents a substrate for idiopathic ventricular arrhythmias [120]. Strauss et. al. (2022) add that predominant right ventricular remodeling promotes multi-wavelet re-entry which underlies ventricular tachycardia [121].

Disorganized GJ distribution and altered anisotropic conduction predispose re-entrant arrhythmias [122]. There is also the implication of endothelin-1 in atrial arrhythmogenesis, and this presents a therapeutic target [123]. However, PAH can be associated with left heart disease, and this is an increasingly prevalent therapeutic problem associated with poor prognosis [124]. This is partly because right ventricular failure induced by blood pressure overload is associated with left ventricular electric remodeling. In addition, reduced Cx43 levels promote this remodeling through impaired cellular impulse transmission [125].

Finally, right heart disease maintains AF due to re-entrant activity, and its underlying substrate involves fibrosis and consequent conduction abnormalities [126].

### Interventions Associated with Connexome Protection in PAH

PAH is characterized by reduced angiotensin-converting enzyme-2 activity (ACE2). In contrast, ACE2 augmentation improves pulmonary hemodynamics and reduces oxidant and inflammatory mediator markers [127]. Targeting this pathway most likely protects the connexome and should prove beneficial in PAH.

There were also the following therapeutic interventions:

The benefit of combined nicorandil and colchicine therapy and associated Cx43 preservation in the PAH rat model [128] and attenuated PAH was recorded in this species due to the propylthiouracil thyroid-suppressing agent’s pleiotropic properties [129].

The bosentan dual endothelin receptor antagonist partly reversed Cx43 remodeling in the PAH hypertrophied right ventricle [130], and combined sildenafil and beraprost inhibited PAH arrhythmogenesis. This combination substantially suppressed hypertrophy and fibrosis and preserved Cx43 expression [131].

Finally, sildenafil confers protection in the PAH rat model by suppressing pro-fibrotic signaling and enhancing Cx43 in the right ventricle [132], and carbenoxolone decreases PAH-induced pulmonary inflammation and arteriolar remodeling in this model by decreasing T-lymphocyte connexin expression.

## 5. Lifestyle Recommendation as Treatment for Arterial Hypertension to Protect Connexome

In the context of the topic of this article it should be emphasized that arterial hypertension and high blood pressure-related cardiovascular disease remain global health hazards posing a major socio-economic and healthcare challenge. The global prevalence of hypertension is estimated to be in the range of 30–45% [38]. In Europe, ∼25% of heart attacks have been linked with hypertension and ∼40% of deaths per annum are caused by hypertension-related cardiovascular disease [133].

Lifestyle interventions are an essential and established part of hypertension management and, in combination with anti-hypertensive drug treatment, provide a most effective strategy to achieve recommended blood pressure targets and reduce cardiovascular mortality. Indeed, guidelines for the management of arterial hypertension strongly recommend lifestyle advice. Mediterranean diet, salt, sugar, and alcohol reduction, smoking cessation, elimination or overcome of stressful events and regular physical activity as well as prevention obesity are essential components not only for the management of hypertension but also for prevention of development of hypertension. 

As suggested by most actual systematic reviews [133,134] non-pharmacological factors are the prerequisite for the analysis of research gaps on the way for the future generation of guideline recommendations on lifestyle treatment in patients with hypertension. Education and routine blood pressure screening should be part of the new perspectives in prevention of hypertension in general population including young. Thus, multi-level approach is a future warranty to reducing the public health burden from increased blood pressure [135] and to protect connexome.

## 6. Conclusions and Perspectives

Research confirms that oxidative stress, a low-grade inflammatory state, and ischemia are involved in hypertension-induced cardiomyocyte junction impairment. Herein we introduce the “connexome” term which combines desmosomes, gap junctions, adherens junctions, and ion channels; and the term is thus defined. The connexome is most important in cardiomyocyte adhesion and the propagation of contractile force, and it is essential for cardiomyocyte coupling which enables electrical and molecular signal transmission. The connexome, therefore, ensures appropriate synchronized heart function; and its impairment is a major component in life-threatening cardiac arrhythmias and heart failure in primary hypertension and pulmonary hypertension. However, there has also been the expressed opinion that preserving the connexome in arterial hypertension treatment can hinder myocardial structural remodeling, hypertrophy, and fibrosis, as well as the occurrence of cardiac arrhythmias.

In conclusion, while major research confirms the beneficial pleiotropic effects of anti-hypertensive and anti-inflammatory agents, further research is essential to identify specific molecular targets and pathways that protect the connexome. This combination will help formulate new clinical approaches to maintain heart function in patients with arterial hypertension. Moreover, lifestyle advice remains one of the crucial pillars of both anti-hypertensive treatment and hypertension prevention.

## Figures and Tables

**Figure 1 biomolecules-13-00330-f001:**
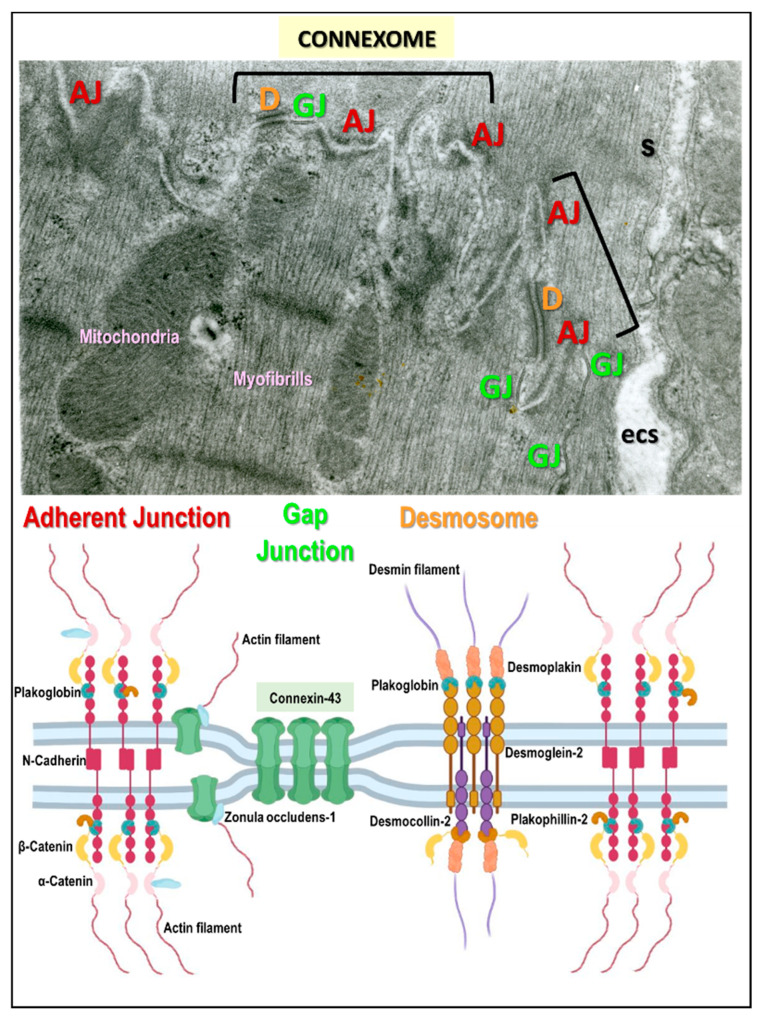
Intercalated discs comprise three distinct junctional complexes, the “connexome”: adherens junction (AJ), desmosome (D), and gap junction (GJ). These work together to mediate cardiomyocyte mechanical and electrical coupling. There are also various interacting proteins that can modulate connexome function [10,12,13,14,15,16,17,18]. Disorders of this complex structure appear to be pro-arrhythmic and promote mechanical heart dysfunction.Electron microscopic images adapted from [11].

**Figure 3 biomolecules-13-00330-f003:**
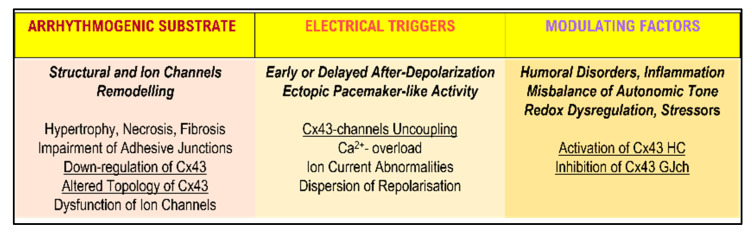
Tentative connexome implications in pro-arrhythmic disorders induced by HTN and PAH.

**Figure 4 biomolecules-13-00330-f004:**
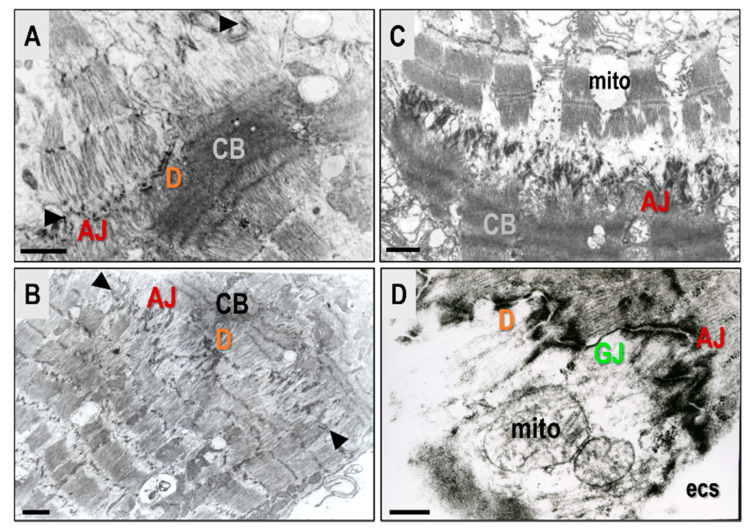
AF and VF aggravate connexome impairment. This includes AJ, D, and GJ due to Ca^2^ -overload, ischemia, and other factors. The subcellular rat heart alterations shown in the (**A**–**C**) images indicate desynchronized contraction of neighboring cardiomyocytes, and image (**D**) shows varying degrees of injury attributed to connexome impairment. Scale bar represents 0.1 micrometer. Adapted from [19].

**Figure 5 biomolecules-13-00330-f005:**
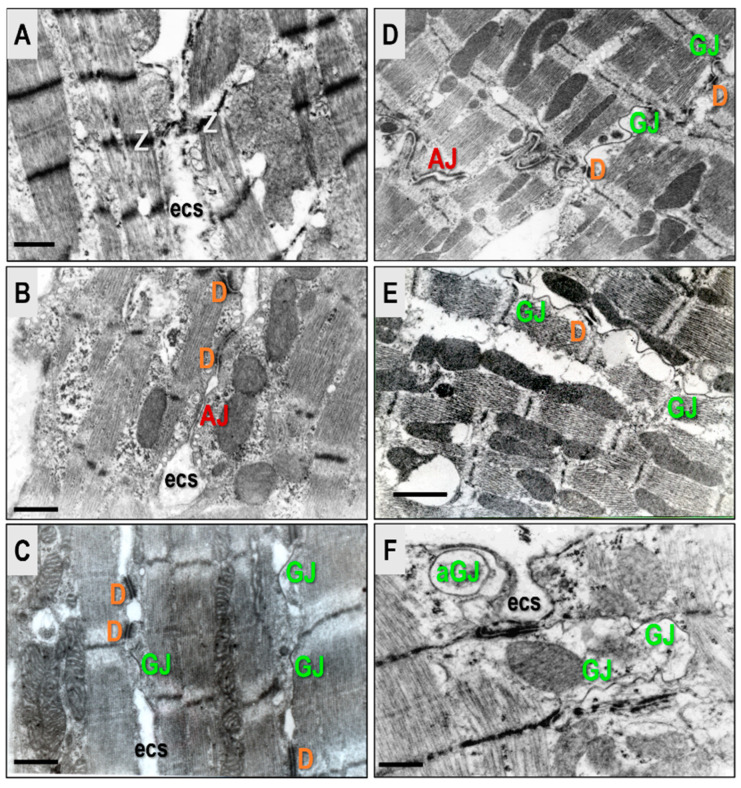
Electron microscope images demonstrate connexome occurrence in SHR hearts. There is formation of the connexome lateral junctions in hypertrophied cardiomyocytes (**A**–**C**), and their presence on the lateral sides with those at the intercalated disc (**D**). A long lateral GJ is obvious in the SHR heart (**E**), and its degradation and the formation of annular profile (aGJ) occur with HTN progression (**F**). Scale bar represents 1 micrometer. Adapted from [54,73].

**Figure 6 biomolecules-13-00330-f006:**
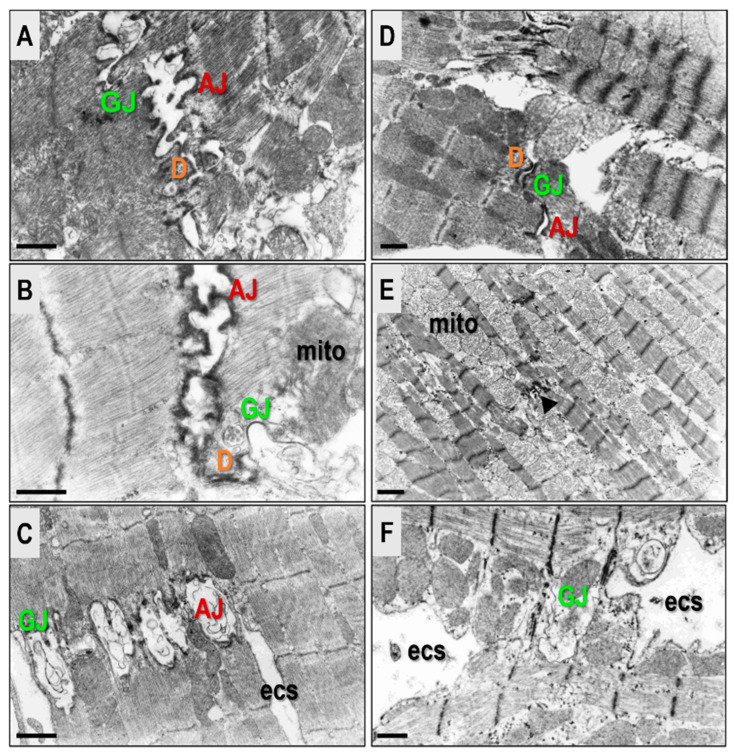
Electron microscope images illustrate connexome impairment in SHR hearts. Varying degrees of AJ dehiscence are apparent in images (**A**–**C**), and these are accompanied by GJs loss. The connexome impairment also results in asynchronized contractions of neighbouring cardiomyocytes (**A**,**D**). Long-lasting HTN related structural remodeling includes severely injured cardiomyocytes, with the destroyed connexome at the intercalated disc (arrowhead in (**E**)) and on the cardiomyocytes lateral side (**F**). This is accompanied with progressive fibrosis. Scale bar represents 1 micrometer. Adapted from [3].

**Figure 7 biomolecules-13-00330-f007:**
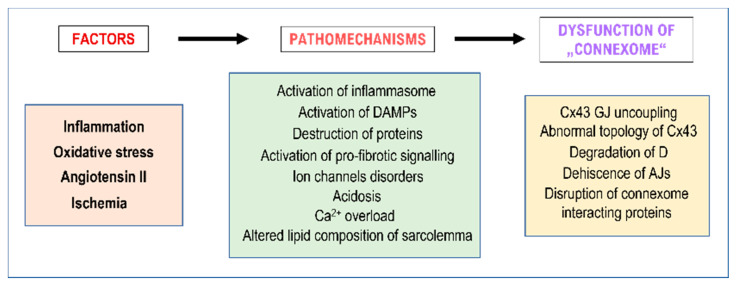
Proposed factors and mechanisms that may be involved in pro-arrhythmic “connexome” dysfunction in chronic arterial hypertension.

## Data Availability

Not applicable.

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
