# Peer review of "Hypertension Induces Pro-arrhythmic Cardiac Connexome Disorders: Protective Effects of Treatment"

_biomolecules, 2023, doi:10.3390/biom13020330_

Round 1

Reviewer 1 Report

The manuscript entitled, ‘Hypertension induces pro-arrhythmic disorders of cardiac connexome: Protective effects of treatment’ is exciting and demanding in the field of cardiovascular sciences. Although this literature review is informative and possesses the reader’s attention toward the development of drugs in HTN and PAH-induced connexome dysfunction and subsequent manifestation of pro-arrhythmic disorders several issues need to be addressed/improved to make it attractive to the readers.

1.      This literature review is missing diagrams/cartoons that represent the possible pathophysiological mechanisms of HTN or PAH-induced connexome dysfunction followed by the development of pro-arrhythmic disorders.

2.      The literature is missing the definition and the role of adherens junction (AJ), desmosome (D), and gap junction (GJ) in the heart or, more specifically, in cardiomyocytes.

3.      Need to define Hypertension and PAH with their pathological characteristics.

4.      On page 1, lines 40-42: The sentence ‘It has been established ………. cardiac arrhythmias’ is missing reference/s

5.      On page 1, lines 42-44: The sentence ‘Available data indicate……………. occurrence of VF or AF’ is missing reference/s.

6.      On page 2, line 47: There is a repetition of ‘at the’.

7.      The writing needs to be improved as there are several typos and grammatical errors throughout the whole manuscript.

Author Response

Thank you very much for your time to review our manuscript. We appreciate your suggestions and comments. Our responses are highlighted by red colour in revised manuscript. English editing was revised by native English speaker.

The manuscript entitled, ‘Hypertension induces pro-arrhythmic disorders of cardiac connexome: Protective effects of treatment’ is exciting and demanding in the field of cardiovascular sciences. Although this literature review is informative and possesses the reader’s attention toward the development of drugs in HTN and PAH-induced connexome dysfunction and subsequent manifestation of pro-arrhythmic disorders several issues need to be addressed/improved to make it attractive to the readers.

We hope that revised manuscript according to your comments will be more attractive for the readers.

  1. This literature review is missing diagrams/cartoons that represent the possible pathophysiological mechanisms of HTN or PAH-induced connexome dysfunction followed by the development of pro-arrhythmic disorders.

Thank you for your suggestion. We added the schema (Figure 7) with proposed factors and mechanisms involved in dysfunction of “connexome” promoting occurrence of cardiac arrhythmias in arterial hypertension.

  1. The literature is missing the definition and the role of adherens junction (AJ), desmosome (D), and gap junction (GJ) in the heart or, more specifically, in cardiomyocytes.

We added following text: Cx43 GJs are essential for electrical coupling and electrical signal (action potential) propagation among cardiomyocytes (Andelova et al. 2021). Fascia adherens junctions (AJs) and desmosomes (D), located mainly in the intercalated disks (ID) are the two types of mechanical junctions responsible for adhesion of cardiac myocytes as well as for mechanical force transduction via actin filaments (Saffitz 2005).

  1. Need to define Hypertension and PAH with their pathological characteristics.

We added short definition and pathology associated with primary HTN and PAH in revised manuscript.

  1. On page 1, lines 40-42: The sentence ‘It has been established ………. cardiac arrhythmias’ is missing reference/s

We added most appropriate reference Danik et al. 2004.       

  1. On page 1, lines 42-44: The sentence ‘Available data indicate……………. occurrence of VF or AF’ is missing reference/s.

We added references Peters et al 1996 and our previous papers Tribulova et al. 2002, 2003, 2009

  1. On page 2, line 47: There is a repetition of ‘at the’.

Sorry for the mistake that we eliminated.

  1. The writing needs to be improved as there are several typos and grammatical errors throughout the whole manuscript.

We apologize very much for typos errors and we hope that English editing by native speaker improved understanding of our manuscript.

Reviewer 2 Report

-This is  a review article describing the role of misregulation of cardiac connexome in HTN and PAH.  Indeed data on this topic are still being collected and comprehensive reviews have been published several years ago. While I was able to see glimpses of the vision of the authors, I cannot be sure that they successfully conveyed that on the paper. Please find bellow my suggestions.

-The review article appears somewhat discordant. Coupled with the need for improvement in syntax, it is difficult to follow in certain parts. For example, lines 141-145: the sentence in line 145 is incomplete. Similarly, lines 159-160 do not make sense. Furthermore, the text jumps from primary HTN, to HF to LVH and then back to HF. Also as a general note, please define all acronyms at the first use in the text. The word "noteworthy" is overused, subsequent sentences are starting with that same word throughout the whole text. 

-Line 140-142: Please rephrase this sentence as it appears unfinished

-Line 166-167: Please define what ZO-1 is. 

Lines 198-200: Please rephrase this entire paragraph. Also please define SHR and whether it is a model that captures human HTN with high fidelity.

-The part about PAH is generally well structured and a bit more easier to follow. 

Author Response

First of all, thank you very much for your time to review our manuscript. We appreciate your suggestions and comments. Responses are highlighted by red colour in revised version. Moreover, the text was edited by native English speaker and the topic may be more attractive for the readers.

This is a review article describing the role of misregulation of cardiac connexome in HTN and PAH.  Indeed, data on this topic are still being collected and comprehensive reviews have been published several years ago. While I was able to see glimpses of the vision of the authors, I cannot be sure that they successfully conveyed that on the paper. Please find bellow my suggestions.

-The review article appears somewhat discordant. Coupled with the need for improvement in syntax, it is difficult to follow in certain parts. For example, lines 141-145: the sentence in line 145 is incomplete. Similarly, lines 159-160 do not make sense. Furthermore, the text jumps from primary HTN, to HF to LVH and then back to HF. Also as a general note, please define all acronyms at the first use in the text. The word "noteworthy" is overused, subsequent sentences are starting with that same word throughout the whole text. 

We did our best to eliminate these mistakes in revised version.

-Line 140-142: Please rephrase this sentence as it appears unfinished

We revised as follows: Subcellular alterations pointing out abnormalities of “connexome,” such as dehiscence of AJs, internalisation and degradation of GJs, are demonstrated on representative images in Figure 5 and Figure 6.

-Line 166-167: Please define what ZO-1 is. 

We revised as follows: Considering that connexin-interacting protein, ZO-1, determines AJs and GJs localization at intercalated disc (Palatinus et al., 2011).

Lines 198-200: Please rephrase this entire paragraph. Also please define SHR and whether it is a model that captures human HTN with high fidelity.

We rephrased this paragraph as follows: The spontaneously hypertensive rat (SHR) is the most frequent experimental model used to imitate human primary HTN. SHR treatment with the phenylbutyrate short-chain fatty acid derivative and captopril both improved myocardial function, regressed cardiac hypertrophy and enhanced recovery from HF (Brooks et al., 2010). It is important here that the gene set combination related to oxidative stress, growth, inflammation, protein degradation and pro-fibrotic TGF-β signaling were downregulated; and these effects were most likely associated with improved connexome function.  

-The part about PAH is generally well structured and a bit easier to follow. 

Thank you once more for your kind approach.

Round 2

Reviewer 1 Report

The authors adequately addressed all the concerns I made. Therefore, I strongly recommend this manuscript to be published at it’s current form.

Reviewer 2 Report

I thank the authors for revising the manuscript. Very well done.